# Integrating Family Farming into School Feeding: A Systematic Review of Challenges and Potential Solutions

Viviany Moura Chaves [1], Cecília Rocha [2], Sávio Marcelino Gomes [3], Michelle Cristine Medeiros Jacob [4,*] and João Bosco Araújo da Costa [5]

1   Postgraduate Program in Social Sciences, Federal University of Rio Grande do Norte, Av. Senador Salgado Filho, s/n, Natal 59078-970, RN, Brazil
2   School of Nutrition and Centre for Studies in Food Security, Toronto Metropolitan University, 350 Victoria Street, Toronto, ON M5B 2K3, Canada
3   Nutrition Department, Federal University of Paraiba, Street Tabelião Stanislau Eloy, s/n, João Pessoa 58050-585, PB, Brazil
4   LabNutrir, Nutrition Department, Federal University of Rio Grande do Norte, Av. Senador Salgado Filho, s/n, Natal 59078-970, RN, Brazil
5   Social Science Department, Federal University of Rio Grande do Norte, Av. Senador Salgado Filho, s/n, Natal 59078-970, RN, Brazil
*   Correspondence: michelle.jacob@ufrn.br; Tel.: +55-84-98189-1234

**Abstract:** Family farming is strengthening its strategic role in school nutrition, but coordinating between school feeding programs and the agricultural sector has proven to be challenging. The goal of this review was to identify the problems that school feeding programs face in acquiring food from family farms. We selected studies from Web of Science, Medline/PubMed, and Scopus and evaluated their methodological quality. Out of 338 studies identified, 37 were considered relevant. We used PRISMA to guide the review process, and we chose not to limit the year or design of the study because it was important to include the largest amount of existing evidence on the topic. We summarized the main conclusions in six categories: local food production, marketing, and logistics channels, legislation, financial costs, communication and coordination, and quality of school menus. In general, the most critical problems emerge from the most fragile point, which is family farming, particularly in the production and support of food, and are influenced by the network of actors, markets, and governments involved. The main problems stem from the lack of investment in family farming and inefficient logistics, which can negatively impact the quality of school meals. Viable solutions include strategies that promote investment in agricultural policies and the organization of family farmers.

**Keywords:** family farming; school feeding; local food chains; food systems

## 1. Introduction

The debate surrounding the transformation of food systems—for greater efficiency, resilience, inclusion, and sustainability—has gained strength in international and national agendas and is a condition for the achievement of the Sustainable Development Goals (SDGs) [1]. Family farming can play an important role in this transformation, particularly in terms of reducing poverty, hunger, and climate change [2,3].

Although there is no universal consensus on the definition of family farming, given its enormous diversity around the world, the Global Action Plan of the United Nations Decade for Family Farming—2019–2028 [2] uses family farming to refer to all models of family-based production—agricultural, forestry, fishing, pastoral, or aquacultural—including peasants, indigenous peoples, traditional communities, fishers, mountain farmers, forest users, and herders. These farms/properties have economic, environmental, social, and cultural functions. It is estimated that globally there are around 570 million family farms,

which occupy 70 to 80% of agricultural land and are responsible for producing 80% of food [4,5]. Evidence shows that family farming, when adequately supported by public policies and investments, has the ability to effectively contribute to food insecurity and poverty reduction [6], biodiversity conservation [7], local economic development [8], and food resilience in times of crisis [9]. Despite their potential benefits, smallholder farmers are most affected by food insecurity and extreme poverty [10]. It has been found that over 70% of people with food insecurity in the world live in rural areas of developing countries, in a disturbing paradox [11]. Many of these people are poorly paid agricultural workers or subsistence farmers who may struggle to meet the food needs of their families.

A shift in the way small producers are viewed has been observed in recent international, national, and regional political debates: they are now seen as central to the resolution of hunger, rather than being viewed simply as part of the problem [12]. Therefore, the issue of how small-scale farming can be crucial to social and environmental protection is of central importance for policy interventions, particularly for school feeding programs.

School feeding programs benefit about 388 million children worldwide, and governments are increasingly recognizing their multiple benefits for populations, such as social protection and food security for students [13]. When linked to family farming, programs can contribute to the development of shorter and closer production chains to schools, while the supply of local and culturally appropriate food can reduce waste and, consequently, carbon emissions [14,15]. From an economic point of view, these programs can also enhance job creation and local economic dynamization, essential factors for reducing poverty and food insecurity in the countryside [16].

However, the growth of initiatives linking family farming and school feeding does not necessarily guarantee their effective implementation in different scenarios. Most of the time, the coordination between the supply of food from family farms and the demand for food for schools can be challenging. Botkins and Roe [17] reported that the challenges that condition the participation of school districts in the Farm to School program in the United States range from high product prices to the unavailability of food throughout the school year. Similarly, some studies [18–21] from Brazil point to the mismatch between the supply of food and school demand, a situation that is directly affected by three main factors: poor logistics, poor communication, and lack of public sector support.

Therefore, our objective with this systematic review is to answer the following question: What are the problems and potential solutions that school feeding programs in different contexts face in acquiring food from family farms? To do this, we aimed to systematize and characterize the evidence produced to date so that future research can fill the identified gaps and contribute to the effective inclusion of family farming in school feeding programs.

## 2. Materials and Methods

This systematic review was conducted based on the PRISMA (Preferred Reporting Items for Systematic Reviews and Meta-Analyses) recommendations. We did not register a protocol for this review because our research does not directly analyze any health-related outcomes [22].

### 2.1. Search Strategy

The research was carried out during the month of March 2022, and three databases were used to conduct the search: Web of Science, Medline/PubMed (via National Library of Medicine), and Scopus, due to their good coverage in collecting evidence for systematic reviews [23]. The research consisted of applying the descriptors, and the search strategies are detailed in the Supplementary Materials.

### 2.2. Eligibility Criteria

We selected articles according to the following eligibility criteria: (i) original articles, published in English, Spanish, or Portuguese; and (ii) studies based on experiences of the participation of family farms in school feeding in different contexts. We chose not to limit

the year or design of the study because it was important to include the largest amount of existing evidence on the topic. We limited our research to papers that included primary data in order to prioritize the most reliable source of information, which helps to reduce bias and increase the validity of the findings.

### 2.3. Studies Selection

A search strategy generated more articles than were eligible according to the eligibility criteria. Screening titles and abstracts in the initial phase allowed for filtering references and eliminating a significant number of studies that did not meet the criteria established for the review. For example, studies that focused on other issues related to school feeding, such as food and nutrition education practices, food safety practices, and estimation of food macro- and micronutrients, were not included if they were not related to family farming. Titles and abstracts were selected based on the descriptors used in the search strategy. We used the Rayyan Qatar Computing Research Institute (QCRI) and Mendeley reference managers to organize the studies based on the merging of the databases and the exclusion of duplicates. Initially, the titles and abstracts were subjected to the first screening by two independent authors (VMC and SMG), in which studies that did not meet the eligibility criteria were excluded. In case of disagreement or uncertainties about inclusion, a third author was consulted (MCMJ). The full texts were retrieved and reviewed by VMC to confirm the eligibility of the study, and in case of doubt, the other authors were consulted. A supplementary manual search was also carried out to identify additional studies based on the references of the selected articles; however, no studies were added.

### 2.4. Data Extraction

Two authors (VMC and SMG) extracted the data and we compiled the relevant information into a spreadsheet for this study. We collected the following information: (i) article data (authors, publication year, journal of publication), (ii) location of the study (city, state, region, and/or country), (iii) overall objective, (iv) type of study, (v) participants, (vi) data collection technique, (vii) variables or categories of analysis, (viii) main results, and (ix) study quality.

We methodologically evaluated the quality of the studies using the following protocols: Consolidated Criteria for Reporting Qualitative Research (COREQ) [24], Joanna Briggs Institute Prevalence Checklist (JBI) [25], Newcastle-Ottawa Scale (NOS) [26], and Analytical Quality Control (AQC) [27]. For the analysis of qualitative studies, we used COREQ, which is a checklist with 32 criteria for evaluation. In the case of observational studies, we considered two protocols, one for cross-sectional studies (JBI) and one for cohort and case-control studies (NOS). Both instruments had nine evaluative items. Finally, we used AQC, a protocol with 21 criteria to analyze the only experimental study in the sample.

After assessing all items, the studies received a score for each criterion met. The quality of the studies was categorized using the criteria of Jacob, Araújo, and Albuquerque [28]. These categories were as follows: strong—when the quality was >80% of the criteria of the referenced checklist; moderate—when it was between 50–80%; and, finally, weak—when it met <50% of the required criteria. In cases of studies with mixed methods and using more than one protocol, we calculated the arithmetic mean and applied it again to the quality categories.

### 2.5. Summary of Results

We present the results descriptively and by absolute frequency. We produced summaries of each of the articles and systematized the main conclusions. The qualitative data was extracted using the thematic analysis technique. These data were segmented, categorized, summarized, and reconstructed to capture the main information that could answer our research question within our data set. In a spreadsheet, the data was organized and analyzed through the following steps: (1) from the main conclusions of the selected studies, we grouped the useful evidence into categories of equal weight; (2) from the evi-

dence, we developed the first level of conclusions (more restricted), highlighting what was common among the studies; and (3) finally, we categorized the restricted conclusions into general conclusions.

## 3. Results

### 3.1. Studies Included

The database search resulted in 338 studies (130 in Web of Science, 44 in Medline/PubMed, and 164 in Scopus). After excluding 137 duplicates, 201 articles were considered eligible for the next screening stage. Based on the titles and abstracts, 66 articles were selected for a full reading. Subsequently, we excluded 29 studies that did not meet the inclusion criteria (*n* = 8) and that were classified as weak in the quality analysis criteria (*n* = 21). Thus, theoretical and review studies, policy briefs, and articles unavailable in full were removed at this stage. In the end, a total of 37 articles were considered eligible and relevant for this systematic review. The process of selecting the articles is described in the flowchart in Figure 1.

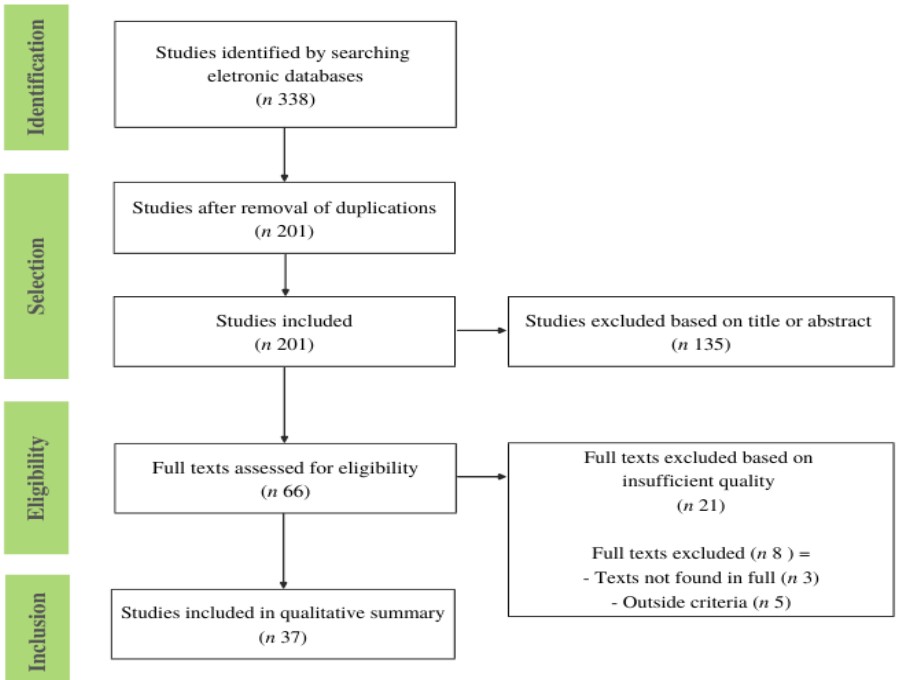

**Figure 1.** Flowchart of the studies selection process.

### 3.2. Studies' Characteristics

Table 1 provides an overview of the main characteristics of the 37 studies in this review. Of the included studies, 27 were published in the last 10 years, 17 of which between 2018 and 2022. The first study in this review was published in 2006 [29], and most of the subsequent publications occurred from 2013 on. Most of the included studies (83.8%) focused on experiences located in the United States and Brazil (see Figure 2). Both countries, as well as the Republic of Ghana, have government programs to include local foods in schools, such as the Farm to School Program (FTS), the National School Feeding Program (PNAE), and the Ghana School Feeding Program. We also found different experiences in European (Spain and Germany) and African (Ghana and Uganda) countries, albeit in small numbers. Some of them refer to the role of agricultural policies or non-governmental organizations in promoting the purchase of local foods for schools. The most commonly adopted study method was observational cross-sectional (54.1%), followed by qualitative studies (27%), cohort studies (10.8%), and mixed methods (8.1%).

**Table 1.** Characterization of studies on family farming in school feeding.

| Study No. | Publication Date (Authors, Year, and Journal) | Setting | Objective | Study-Type | Participants | Data Collection Technique | Variables/Categories Analyzed | Important Outcomes | Quality |
|---|---|---|---|---|---|---|---|---|---|
| 1 | Baccarin et al., 2017, Revista de Economia e Sociologia Rural [30] | São Paulo, Paraná, and Santa Catarina, Brazil | Standardize indicators to evaluate the purchases from family farming for school feeding. | Cross-sectional | Not applicable | Documentary research—bidding processes, supplier contracts, and rendering of accounts | Number of bidding processes, quantify of products of animal and vegetable origin, degrees of processing, delivery frequency, number of active farmers, number of receiving units. | Properly conducted bidding processes can favor the participation of farmers. Lack of information and transparency hinders the organization of family farmers and the efficiency of the process. | Moderate |
| 2 | Bonanno, Mendis, 2021, Food Policy [31] | United States | Understand the factors that are associated with the repeated participation of school districts in the Farm to School Program. | Cohort | 6798 school districts | Secondary Data—Farm to School Census, 2013–2017 | Size of schools, % of students eligible for school meals, cost of food, race. Specific variables about Farm to School and state policies. | School districts that participate in the Farm to School, when they prefer local food procurement, are about 5% more likely to stay in the program. The number of implemented activities is positively associated with the decision to remain in the program. | Strong |
| 3 | Botkins, Brian, 2018, Food Policy [17] | United States | Analyze the factors associated with the decision of school districts to participate in the Farm to School Program. | Cross-sectional | 9.643 school districts | Secondary Data—Farm to School Census, 2013 | Supply data, food environment, school characteristics, location, and race. | Schools in areas with more districts that have already implemented programs tend to have fewer difficulties in implementing and executing the program, and the probability of participation increases (spillover effect). | Strong |
| 4 | Braun et al., 2018, Sustainability [32] | Berlin, Germany | Investigate the value chain in providing organic food to schools. | Case study. Qualitative | 14 actors at different levels of the supply chain | Interviews | Internal structures: actors' perception, values, and attitudes towards the local/organic; practices in the value chain: marketing, purchasing, collaboration; context analysis; structural factors (price, purchasing policy, local food demand). | Although purchasing policies have increased the availability of organic foods in school meals, the value chains for locally produced foods are limited and undervalued (lack of incentives and resources). | Moderate |
| 5 | Carvalho, Oliveira and Silva, 2014, Interface—Botucatu [33] | Cidade do Bonfim, Bahia, Brazil | Analyze the symbolic and social perceptions of quilombolas, in the offer of agricultural food, to the National School Feeding Program, to promote food and nutritional security. | Ethnography, qualitative | 14 actors interviewed | Participant observation and interview | Land regulation provides conditions for permanence, income generation and belonging. At the same time, young people are disenchanted with rural work. School feeding enabled the reduction of hunger and the cultural perpetuation of culinary practices and local foods. | The community conceives and values the "natural" food from the land as a source of survival and local development, seeing in school feeding an opportunity to guarantee food security. | Moderate |
| 6 | Castellani et al., 2017, Revista de Nutrição [18] | Santa Catarina, Brazil | Describe purchases of food from family farms and organic foods by the National School Feeding Program. | Cross-sectional | Nutritionists and education secretaries from 293 municipalities | Questionnaire | Acquisition of family farming products and organic food, purchase percentage, difficulties in purchasing organic products. Data such as the size of municipalities, Municipal Human Development Index (MHDI), and number of students were also associated. | Half of the municipalities purchased organic food for school meals, and a third did not reach the minimum percentage required for family farming purchases. Those with the lowest MHDI and the lowest number of students had more difficulty both in the acquisition and organic products. | Strong |
| 7 | Christensen et al., 2017, Renewable Agriculture and Food Systems [34] | United States | Analyze how school districts source local foods and the relationship to local non-dairy food expenditures per student. | Cross-sectional | 2.689 school districts | Secondary Data—Farm to School Census, 2015 | Supply chains used by school districts for local food purchases, the size of school districts, types of products purchased, and % of students getting free meals. | Schools that purchase local foods from traditional distributors are likely to have higher average spending per student compared with schools that purchase local foods directly from farmers or non-traditional distributors. | Strong |
| 8 | Colasanti et al., 2012, Journal of Nutrition Education and Behavior [35] | Michigan, United States | Investigate changes in the perspective of school food-service directors in Michigan in a 2004 survey and the factors facilitating the expansion of the Farm to School. | Cross-sectional | 270 food service directors from the U.S. Department of Agriculture's National School Feeding Program | Questionnaire and interviews | Behaviors, interests, motivations, concerns, and barriers in purchasing local food for schools. | Participation in Farm to School was 3 times higher than in 2004. Food service directors' motivations for buying local food were as follows: supporting producers, ensuring better quality food, and supporting the local economy. | Strong |
| 9 | Silverio, De Sousa, 2014, Revista de Nutrição [36] | Santa Catarina, Brazil | Analyze suggestions from social actors of school feeding in municipalities of Santa Catarina to facilitate the use of organic food from family farming. | Qualitative, exploratory | 1st stage: 293 municipalities; 2nd stage: 52 municipalities | Questionnaire and a semi-structured interview with relevant social actors | 684 suggestions from 446 social actors were identified. Changes in logistics, quality control, menu planning, government incentives, and methods of encouraging the consumption of organic foods were suggested. | Farmers suggested less bureaucracy and outsourcing, fewer taxes and more management involvement. Problems were identified with the supply of specific foods, low diversity of organic foods, and lack of certification | Moderate |

**Table 1.** *Cont.*

| Study No. | Publication Date (Authors, Year, and Journal | Setting | Objective | Study-Type | Participants | Data Collection Technique | Variables/Categories Analyzed | Important Outcomes | Quality |
|---|---|---|---|---|---|---|---|---|---|
| 10 | De Souza, Villar, 2019, Revista de Nutrição [37] | São Paulo, Brazil | Describe and analyze aspects of implementing the purchase of food from family farming in the National School Feeding Program. | Cross-sectional, descriptive-analytical | 25 municipalities and 105 schools | Questionnaire | Percentage of purchase, type of management, number of students, number of schools, size of the municipality, aspects of implementing purchases from family farming. | The type of management of the PNAE and characteristics such as the size of the municipality, number of students, and public schools, can influence the implementation of the purchase of food from family farming for schools. | Strong |
| 11 | Dos Anjos, Lopes Filho, Horta, 2022, Ciência Rural [38] | Minas Gerais, Brazil | To identify sociodemographic, economic, and agricultural characteristics and associate them with compliance with the 30% requirement in municipalities in Minas Gerais in 2017 | Cross-sectional, descriptive-analytical | 848 municipalities | Secondary data | Purchase percentage, per capita, number of inhabitants, territorial area, number of students, Municipal Human Development Index (MHDI), agricultural data. | Evidence associates characteristics of agricultural management (such as policies to support family farming) with meeting the goal of acquiring food. | Strong |
| 12 | Elolu, Ongeng, 2020, BMC Nutrition [39] | Uganda | Examine the feasibility of a community-based action research to empower rural food vendors to use local foods to produce nutritionally enhanced products for school feeding. | Mixed-method action research and experimental study | 1st phase: women food vendors, school administrators and teachers, and community members (parents). 2nd phase: 180 students between 10 and 14 years old. | Focus group, analysis of nutritional composition (macronutrients and micronutrients), sensory and acceptability evaluation | Perceptions of school feeding—local alternatives (community gardens), improvement of local food resources, community-level partnerships, nutritional interventions for local application. Correlation between the nutritional composition of original and improved gari (cassava-based product). Application of hedonic scale and sensory attributes. | Community-sensitive nutrition innovation provided alternatives for rural vendors to address schoolchildren's hunger. The action research resulted in a highly accepted, nutritionally improved product with superior nutritional properties. | Moderate |
| 13 | Ferreira et al., 2019, Revista de Saúde pública [21] | Rio de Janeiro, Brazil | Identify the perception of operating agents in the National School Feeding Program. | Cross-sectional | 100 program agents from 38 municipalities | Semi-structured questionnaire | Percentage of purchases and types of food from family farming, difficulties encountered in implementing the program, educational activities, and performance of the School Feeding Council. | Need to hire nutritionists to meet the demands of the PNAE; investment in educational activities on healthy eating in schools; training of counselors in School Feeding; and assistance to family farmers to facilitate participation and diversify food. | Moderate |
| 14 | Fitzsimmons, O'Hara, 2019, Agricultural and Resource Economics Review [40] | United States | Verify whether market channel procurement strategies for local food affect schools' perceptions and whether meal costs decrease as a result of participation in the Farm to School. | Cross-sectional | 2102 school districts | Secondary Data—Farm to School Census, 2013 and 2015 | Location, school, implementation, acquisition challenges, food environment, acquisition strategy. | They found that market channel acquisition strategies can contribute to reducing school lunch costs. The probability of schools obtaining local food from intermediaries is influenced by the number of direct marketing producers in the municipality. | Strong |
| 15 | Giombi et al., 2020, Journal of Agriculture, Food Systems, and Community Development [41] | Oregon, United States | Evaluate the opt-in acquisition feature (name of feature) of the Oregon Farm to School grant program during the 2015–2016 period. | Cohort | 212 school districts and 1485 schools | Secondary data—Oregon Department of Education 2014–2015 (baseline) and 2015–2015 (intervention) | Demographics—district size, district income status, % of non-white students, % of students eligible for free or discounted lunch. Key local shopping data—total food expenditure, fruit and vegetable expenditure. | The opt-in approach to the grant program facilitated greater participation from low-income districts that otherwise would not have accessed the grant program. Under the opt-in program, 89% of children eligible for free and reduced-price meals attended schools in participating districts compared with 39% of children eligible under the competitive program. | Strong |
| 16 | Greer et al., 2018, Journal of School Health [42] | Bridgeport, Connecticut, United States | Examine opportunities to promote local products and consumption among high school students in an ethnically diverse, low-income urban community. | Qualitative | 53 students from 3 high schools | Focus group and questionnaire | Students' understanding of locally produced products, their benefits, and the quality of food; costs of consuming local products and the importance of promoting them. | Students concluded that local products were of higher quality than non-local ones. Students also pointed out that the consumption of local food is associated with care for the environment. | Moderate |
| 17 | Izumi, Wright, Hamm, 2010, Journal of Rural Studies [43] | In the Midwest and Northeast regions of the United States | Evaluate the motivations of farmers who participate in Farm to School programs. | Qualitative | School districts with participation from farmers, school feeding service professionals, and food distributors | Semi-structured interviews and data from menus, requests for proposals, price lists, and other documents | Quantitative data: farm size, produce grown/animals raised, packaging facility, outlets, % sales for Farm to School. Qualitative data: strategies adopted by farmers, social and environmental benefits. | Findings suggest that farmers sold their products to schools for two main reasons: diversifying their marketing strategies and contributing to social benefits through direct actions. | Moderate |

**Table 1.** *Cont.*

| Study No. | Publication Date (Authors, Year, and Journal | Setting | Objective | Study-Type | Participants | Data Collection Technique | Variables/Categories Analyzed | Important Outcomes | Quality |
|---|---|---|---|---|---|---|---|---|---|
| 18 | Izumi, Alaimo, Hamm, 2010, Journal of Nutrition Education and Behavior [44] | In the Upper Midwest and Northeast regions of the United States | Explore the potential of Farm to School programs to simultaneously improve children's diets and provide viable market opportunities for farmers. | Qualitative, case study | 18 participants (school food service professionals, farmers, and food distributors) | Interviews and documentary research | General characterization of the districts that participated in Farm to School—location, population, free and reduced lunch participation rate, distribution strategy, student perception, food quality, interaction with farmers, food prices. | Students' preference for locally grown food was related to food quality, the influence of school staff, and relationships with farmers. Buying food directly from farmers and wholesalers was associated with lower prices and flexible specifications and "feeling local." | Moderate |
| 19 | Izumi et al., 2006, Journal of School Health [29] | Michigan, United States | Investigate the interest of Michigan school food service directors and the opportunities and barriers to implementing a Farm to School program | Cross-sectional | 664 food service directors representing school districts | Questionnaire | Degree of interest in obtaining local foods, motivation to serve local foods at school, most purchased foods, concerns about purchasing local foods, limitations that prevent buying directly from local producers. | Main interests in school farming: supporting the local economy and community, access to better quality food, and encouraging the consumption of fruits and vegetables by students. Reported concerns included cost, federal and state purchasing regulations, reliable supply, fruit and vegetable seasonality, and food safety. | Strong |
| 20 | Lehnerd et al., 2018, Journal of Agriculture, Food Systems, and Community Development [45] | Pennsylvania, Maryland, Virginia, West Virginia, New Jersey, Delaware, and Washington DC, United States | Explore farmers' perceptions, barriers to adoption, and impacts of the Nutrition Incentive, and Farm to School programs. | Cross-sectional | 155 farmers | Questionnaire | Farm and farmer characteristics—location, income, farm size, ownership status, sales, marketing, or production management, age, and years of cultivation. | Farmers have realized that both programs provide beneficial social impact and economic opportunities. The most significant barriers relate to issues with product pricing, customer engagement, and logistics. | Strong |
| 21 | Long et al., 2021, The Journal of school health [46] | Colorado, United States | Evaluate how the number of local fruits and vegetables purchased in 3 northern Colorado school districts might change in response to a statewide policy that provides reimbursements for food purchases. | Cohort | 3 school districts | Receipts for food purchases for schools | Socioeconomic characteristics of schools and districts. | An optimization model was built that mimics the decisions made by Food Services Directors. The results of this optimization model reveal that local food purchases can increase by 11–12% in response to a Colorado policy that provides a refund of USD 0.05 per meal for local food purchases. | Strong |
| 22 | McCarthy, Houser, 2017, Journal of Hunger and Environmental Nutrition [47] | United States | Determine if school districts in states with local food laws for schools have significantly higher participation in Farm to School programs and if they serve local food more often compared with districts in states without laws. | Cross-sectional | 9.887 school districts | Secondary data—2011–2012 Farm to School Census | Presence of law, presence of the Farm to School program, frequency of serving local foods according to 12 food groups, other factors associated with school districts (number of students, location, region). | The presence of local food-related laws was associated with a greater likelihood of having Farm to School programs and serving local foods at higher frequencies in school feeding programs. | Strong |
| 23 | O'Hara, Benson, 2017, Renewable Agriculture and Food Systems [48] | United States | Estimate the responsiveness of local food supply by schools in response to changes in local agricultural production. | Cross-sectional | 12.585 school feeding authorities | National secondary data—2015 | School district expenses for local non-dairy products; school district spending on local dairy products; agricultural direct-to-consumer sales within 100 miles of the district; per capita income. | Increasing local agricultural production increases the likelihood that schools will purchase local produce. Poorer schools and schools in poorer counties are less likely to purchase food locally. | Strong |
| 24 | O'Hara, McClenachan, 2019, Marine Policy [49] | United States | Identify attributes that influence school purchases of local seafood at both the US school and regional level. | Cross-sectional | 4719 school-feeding authorities with farm to school initiatives | National secondary data—2013–2015 | Characteristics of seafood served (types and quantity) and characteristics of school districts participating in school feeding programs (number of students, income, etc.). | Three factors emerged as strong influences on local seafood procurement: proximity to seafood ports, outreach and promotion efforts, and the geographic region of the school feeding program. | Strong |
| 25 | Pinard et al., 2013, Preventing Chronic Disease [50] | Douglas County, Nebraska, United States | Assess the feasibility, interest, and barriers to implementing farm to school activities in 7 school districts. | Cohort | 7 directors of the school food service, 5 distributors, and 57 local producers | Questionnaire | This research evaluated: school meal programs, facility capacity, food purchasing, local food practices, and barriers to offering these foods. In terms of the producer, it evaluated the type of production, sales practices, and willingness to participate in the program. As for the distributor, it evaluated distribution and service area practices, sales to schools, and willingness to participate in the Farm to School program. | The participation of school feeding services in Farm to School improved after the intervention, showing increases in interest in purchasing local foods. They reported difficulty in finding farmers to purchase from with (1) food safety standards, (2) inability to provide throughout the school year, (3) inability to produce enough volume of product, and (4) with more competitive prices. | Moderate |

**Table 1.** *Cont.*

| Study No. | Publication Date (Authors, Year, and Journal | Setting | Objective | Study-Type | Participants | Data Collection Technique | Variables/Categories Analyzed | Important Outcomes | Quality |
|---|---|---|---|---|---|---|---|---|---|
| 26 | Rockett et al., 2019, Ciência Rural [51] | Rio Grande do Sul, Brazil | Examine the profile of family farming food acquisition for schools in municipalities of Rio Grande do Sul. | Cross-sectional | 371 municipalities | Questionnaire | Data on resource management, percentage of acquisition of food from family farming for school feeding, criteria considered in the development of food menus, social actors involved, financial resources used, challenges and obstacles faced, the origin of purchases, delivery, and socio-biodiversity products were analyzed. | Only 1.1% of the municipalities did not purchase from family farming. Nutritionists, farmers, and formal organizations had the highest participation in the process. The main challenge cited was the disorganization of farmers, low capacity to meet demand, and the variety of products. | Strong |
| 27 | Plakias, Klaiber, Roe, 2020, Journal of Agricultural and Resource Economics [52] | United States | Investigate the relationship between local food storage capacity and length of the local food supply chain with local food expenditure in districts. | Cross-sectional | 4242 school districts | National school-based study using questionnaires | Local food expenses, number of students in the district, free lunch, local food storage area, number of farm to school policies enacted in the state, purchase of local fruit, vegetables, and meat. | Increasing the food spillover radius by 50 miles and obtaining intermediaries increase the average district's local spending by 8% and 26%, respectively. District actions increase student access to local foods by expanding local definitions or obtaining through intermediaries, and, therefore, have the potential to reduce localized benefits for nearby farmers and community members. | Strong |
| 28 | Schafft, Hinrichs, Bloom, 2010, Journal of Hunger & Environmental Nutrition [53] | Pennsylvania, United States | Examine the current forms, organization, and needs of the Farm to School program policy in the state of Pennsylvania. | Cross-sectional, quantitative, and qualitative | 378 school food services directors | Questionnaire | The interviewee's familiarity with the Farm to School program, opinions on the benefits and challenges of local food purchasing, food purchasing practices, and the structure and capacity of the school district's food service were assessed. | Only 10% of directors reported familiarity with the program. The challenges for local food purchasing were as follows: seasonal availability of local fruits and vegetables, inadequate supply, and inconsistency in local foods' quality and delivery. | Moderate |
| 29 | Shaibu, Al-Hassan, 2015, Agris Online Papers in Economics and Informatics [54] | Tamale, Tolon-Kumbungu and Karaga, Northern Region of Ghana | Analyze the accessibility of rice producers to Ghana's school feeding program and its effect on production in three districts. | Cross-sectional | 100 rice farmers | Questionnaire | The dependent variable was the amount of rice production, and the independent variables were the use of pesticides, agricultural labor, total farm size, farmer age, application of fertilizers, extension visits, and access to the school feeding program. | Agricultural labor, farm size, and applied fertilizer were important variables in increasing production among farmers, while accessing the market through Ghana's school feeding program was not significant. | Strong |
| 30 | Shaibu, Al-Hassan, 2014, Agris Online Papers in Economics and Informatics [55] | Tamale, Tolon-Kumbungu and Karaga, Northern Region of Ghana | Evaluate the factors that influence suppliers of the Ghana School Feeding Program to purchase rice from local farmers. | Cross-sectional | 100 rice farmers and 50 suppliers of the school feeding program | Questionnaire | The processing cost, distance from the supplier to the school, availability of storage facilities, price of milled rice, the student population at the school, and the ease with which the supplier can locate the rice farmer were considered. | Factors that influenced suppliers include the availability of storage facilities, other work carried out by the suppliers, price of rice, ease of locating rice producers, and delays in payment for food vouchers. Payment and availability of storage are essential to addressing the issue of purchasing directly from farmers. | Strong |
| 31 | Smith et al., 2013, Childhood obesity [56] | Southern Illinois, United States | Investigate the influence that school size has on the perceptions of elementary and secondary school food service staff on the benefits and barriers, and their attitudes towards purchasing local food. | Cross-sectional | 78 food service buyers and 62 school food services employees | Questionnaire | The benefits of buying local foods, obstacles to buying local foods, attitudes towards buying more local foods, and information about the school food service program was evaluated. | Buyers consider the seasonality, volume, quality, and safety of local foods as challenges to buying. The benefits of buying local foods are better in large schools compared with small and medium ones. | Strong |
| 32 | Soares et al., 2021, International Journal of Environmental Research and Public Health [57] | Spain | Explore the facilitating factors and opportunities that can promote the implementation of local food purchases in the Spanish school lunch program according to key informants. | Qualitative | 14 participants: consumer/producer organizations; buy-local supporters for government schools, academics | Interviews | The analysis of the interviews resulted in five categories: social fabric, policy, public agenda, regional characteristics, and regional context. These categories were divided into 14 subcategories. | Overlap between social and political demands was seen as facilitating factors for purchasing local food. The presence of health and sustainability issues on the public agenda, the existence of a structured production system, and political changes represent an opportunity to implement these purchases. | Moderate |

**Table 1.** *Cont.*

| Study No. | Publication Date (Authors, Year, and Journal | Setting | Objective | Study-Type | Participants | Data Collection Technique | Variables/Categories Analyzed | Important Outcomes | Quality |
|---|---|---|---|---|---|---|---|---|---|
| 33 | Soares, Caballero, Davó-Blanes, 2017, *Gaceta Sanitaria* [58] | Andalucía, Canarias and Asturias, Spain | Explore and compare the characteristics of primary education centers in terms of purchasing local food for school meals. | Cross-sectional | 139 directors of primary education centers and 47 other professionals linked to teaching, and administrative sectors responsible for canteens | Online questionnaire | Location (urban/rural), center size, number of meal-serving students, percentage of students receiving free meals, the average cost of the school menu, presence of a healthy meal program, food management responsibility, infrastructure, purchase of organic products, benefits, difficulties, and purchased products. | Primary education centers in rural areas are more likely to purchase local food. Centers with a lower average cost of the school menu purchase more local food, and most of them have healthy eating programs, self-manage their diners and their kitchen, and purchase more organic food. Most schools that purchase local food identify the productive capacity of the region as a challenge, as well as seasonal variation and lack of support from public institutions. | Moderate |
| 34 | Soares et al., 2017, British Food Journal [59] | Santa Catarina, Brazil | Investigate the effect of using food products from family farming on the school menu of the School Feeding Program in a municipality in southern Brazil. | Cross-sectional Quantitative and qualitative | 16 key informants involved in the school feeding program or agricultural production | Interviews and document analysis | School menus, purchase and sale of food products, menu composition, distribution and transportation, quantity and quality of products from family farms, production method, and support from other food programs. | The direct supply of local family farming food resulted in an improvement in the school feeding program of the municipality. Along with an increase in the quantity and variety of fresh and organic foods, there was a reduction in the supply of industrialized foods. | Moderate |
| 35 | Soares et al., 2015, Ciência & Saúde Coletiva [60] | Santa Catarina, Brazil | Evaluate the compliance with the recommendations of the school feeding program for the acquisition of products from family farming. | Qualitative | 7 managers and employees of the school feeding program, 5 managers and employees of the secretary of agriculture, and 4 representatives of the farmers' organization | Semi-structured interviews | Characterization of school meals, menu planning, supplier selection, purchasing system, product receipt, activities carried out by counselors, characterization of agriculture in the municipality, production planning, supplier selection, delivery of products, characterization of farmers participating in the cooperative. | It was identified that the delivery of products and meeting demand were being carried out in accordance with recommendations. Non-conformities were identified in the preparation of the public call and sales project, as well as in the adherence to product quality standards. It was observed that regular food supply was facilitated by the diversity of suppliers and the exchange of food between the cooperative and neighboring municipalities. | Moderate |
| 36 | Thompson, Brawner, Kaila, 2017, Agriculture and Human Values [61] | Georgia, United States | Investigate the perceptions of food security as an emerging barrier in farm to school efforts to bring local food to schools. | Ethnographic study, qualitative | 17 program operator agents | Semi-structured interviews | Food safety data (hygienic-sanitary quality risk) | Program agents resort to purchasing through national supply chains or local producers who sell through conventional channels due to issues with the safety of local food, despite their ideology of supporting the purchase from small local farmers. | Moderate |
| 37 | Virta, Love, 2020, Health Behavior And Policy Review [62] | Oregon, United States | Identify how Fish to School Programs are implemented, their impacts, and the factors that enable support for these programs | Formative research, qualitative | 2 school districts, 6 interviews | Exploratory interviews. Respondents included seafood processors, Oregon Seafood Commission leaders, school district food service leaders, and school kitchen managers. | Information about fish species, quantity, price, percentage increase compared with the base price, and the total cost was collected. General characteristics of the school districts such as the number of students and schools, race, and percentage of the population below the poverty line was also considered. | The factors that facilitated the School to Fish programs included strong program leaders and partnerships, funding from Farm to School subsidies, and creative use of resources. Challenges in maintaining the program included sustainably funding the program, seafood distribution networks, recipe development, and higher cost per portion of seafood compared with other proteins. | Moderate |

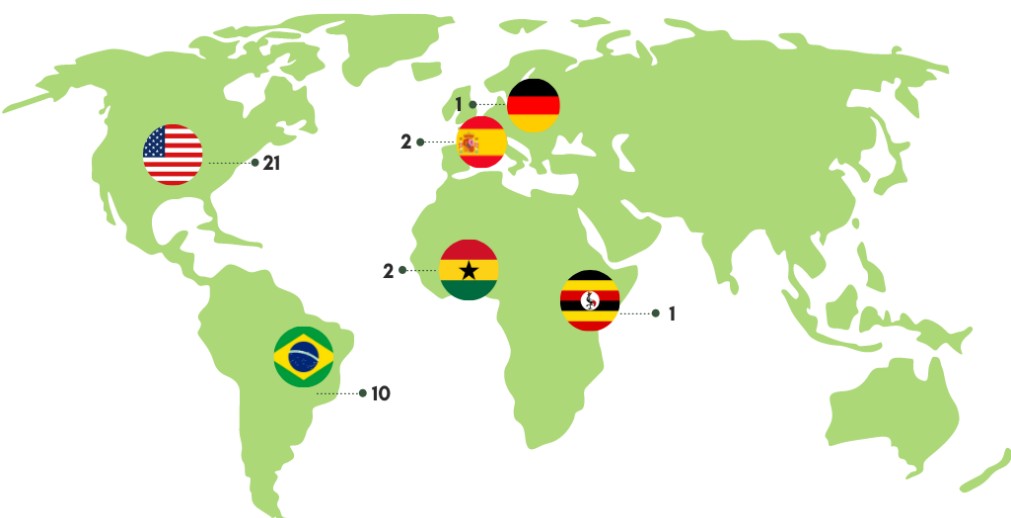

**Figure 2.** Number of studies distributed by countries.

### 3.3. Quality Analysis

We selected studies with strong (*n* = 20) and moderate (*n* = 17) quality. Studies classified as low quality were generally qualitative in methodology, followed by cross-sectional and cohort studies. For qualitative studies, the omission of relevant information about the methodological design and data analysis weighed on the decision, as well as the consistency between the presented data and conclusions. In both cross-sectional and cohort studies, the main weaknesses were the limitations of small sample size and lack of representativeness, as well as issues with sampling criteria and variable measurement.

### 3.4. Difficulties in Acquisition of Food from Family Farming for School Feeding

This study sought to identify the challenges faced by school feeding programs in the acquisition of food from family farms. Based on the main findings of this review, we present six categories that directly and indirectly influence the acquisition of family-based foods for schools, grouped into three dimensions corresponding to food production, acquisition support elements, and consumption (Figure 3). In relation to food production, we highlighted the impacts that affect the supply of food to schools, from productive capacity to commercialization logistics and market competition between small, medium, and large producers. In the second dimension, we listed studies that report difficulties related to support elements—those that guide and regulate the dynamics of food acquisitions in school feeding programs—such as the current legislation (local, regional, or national), costs and expenses generated by foods, and communication/articulation among the stakeholders involved. Finally, we highlighted the barriers involving the consumption of food in schools, considering aspects related to health, culture, and the environment.

#### 3.4.1. Production

We found that low production capacity, irregular supply, and unpredictability in food supply [29,50,53,58] are barriers reinforced by the social, economic, environmental, and political characteristics of the locations [30,38,49,56,59]. In general, municipalities or school districts that are less likely to adopt local food purchases are poorer and have a lower Human Development Index (HDI) [18,37,48]. In addition to this, the size of the municipality/school district and the distance between production and schools are factors that influence access to these foods. While large municipalities are more attractive to farmers due to the high demand generated, some studies have concluded that the larger the municipality, the greater the logistical difficulties in supplying, transporting, and storing food [18,37]. The same situation was observed in school districts. Smaller districts had fewer obstacles (for example, delivery costs, supplier payments, and food volume)

compared with medium and large districts [56]. With regard to the distance between production and schools, some studies showed the influence of distance on production costs, delivery dynamics, diversity, and quality of the foods offered [49,55,59]. Soares et al. [59] reported in a study conducted in southern Brazil that proximity between production and school reduces transportation time and favors the consumption of fresher and healthier foods.

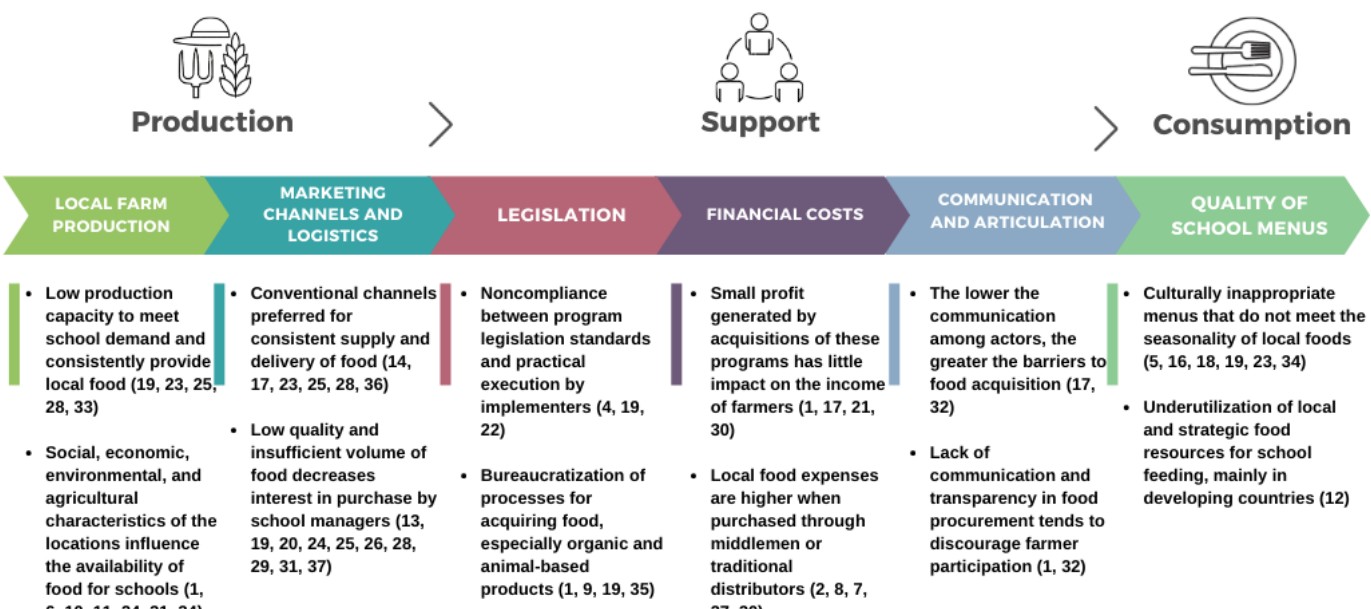

**Figure 3.** Summary of the main conclusions of studies that analyzed the participation of family farming in school feeding. Numbers corresponding to Table 1 (see column 1 "Study no").

Findings regarding the commercialization of food for schools demonstrate that logistical challenges—including irregular supply and low production capacity—in models of local food acquisition directly from the producer increasingly reinforce the participation of national food chains and intermediaries in this market niche [40,43,49,50,53,61]. Some studies justify that the benefits brought by these distribution and commercialization channels include the ability to provide food throughout the school year, in sufficient volume, at competitive prices, and to the required sanitary standards [21,29,40,45,51,54,56,62]. On the other hand, while the efficiency of these channels in regularly supplying food is recognized, the social, cultural, economic, and environmental impacts of these foods traveling long distances through complex supply chains are also questioned [43].

3.4.2. Support

Our review suggests that contradiction in the regulatory devices of the programs tends to produce new conflicts between those who should benefit. Studies have reported that policies/programs with contradictory and inconsistent laws create room for broad interpretations by their implementers and therefore run the risk of not being adequately carried out [29,32,47]. Braun et al. [32] found that while implementing a policy of purchasing organic and locally sourced foods for school meals in Berlin led to an increase in the availability of organic foods in schools, the origin of these foods was not local. As a result, the new policy provided little benefit to the local farming niche, which did not see increased demand following its implementation. At the same time, the excess of regulation increases the bureaucratization of purchases and limits the power of sale of these foods [29,30,36,60]. In the case of Brazil, for example, the bureaucracy of public purchases mainly affects the sanitary standards for the acquisition of animal products [38,51], the certification of organic foods [18,36], and the process of how the foods are acquired (through administrative instruments) [30,60]. In these circumstances, most family farmers, when unable to meet the

requirements necessary to participate in this market, are discouraged and conditioned to sell their products at local markets, since marketing for government programs runs into regulatory devices.

In regards to financial costs, while school agriculture programs have been widely promoted as a market opportunity for family farming, the profit generated by these acquisitions has little impact on farmers' income [43]. In general, the profits represent a very small portion of the total agricultural income of the farmer, which is mainly composed of sales at farmer's markets and to intermediaries [30,43]. Part of these payments is delayed and further discourages the participation of farmers [55,56]. The food costs related to school purchases have also been classified as a barrier to the maintenance of these programs [31,35,46]. It has been suggested in studies that when schools purchase local food from conventional distributors, the financial costs may be higher compared to schools purchasing directly from farmers [32,36,41,47]. In addition, instability in food prices can also dictate the dynamics of acquisitions, so when prices rise, the supply and demand opportunities for the programs decrease [17].

Our findings also highlight that some of the difficulties encountered in purchasing food for schools are related to weak communication among the network of actors [30,43,57]. In these cases, the lack of communication, coordination, and transparency are recurring issues in different scenarios that tend to distance the inclusion of farmers in these programs [30,57]. As reported in the study by Izumi et al. [43], when the number of nodes between producers and school food agents increases, the opportunity for coordination between the ends of the chain is lost, and all actors are disadvantaged.

### 3.4.3. Consumption

Studies show that the absence of locally grown foods negatively impacts the quality of school meals, making them less healthy and culturally inappropriate [33,42,48]. We also found that the inclusion of local foods was associated with higher consumption of fruits and vegetables and lower consumption of processed foods among students, indicating a positive relationship between locally grown foods and fruit and vegetable consumption [18,48]. From a sensory perspective, Greer et al. [42] found in their research that, in the perception of students, local products were of higher quality than non-local products (taste and freshness). Students also pointed out that the consumption of local foods is associated with caring for the environment. From a cultural standpoint, school menus based on local food preparations are associated with greater acceptance by the school community, as well as contributing to the preservation of food habits, biodiversity conservation, and sustainability [33]. In addition, the underutilization of locally available food resources and lack of knowledge about their food potential reduce opportunities for the inclusion of strategic foods in school meals, especially in economically disadvantaged areas [39].

## 4. Discussion

The main objective of this review was to identify difficulties faced by initiatives in school feeding programs to acquire local food from family farms. Based on our analysis, we saw that although experiences linking local food to schools have expanded worldwide in recent years, this growth is not necessarily reflected in the consolidation of these initiatives. In general, difficulties occur at all stages of acquisition, from production to consumption, and are influenced by the network of actors, markets, and governments involved. However, the studies analyzed indicate that the most critical problems emerge from the most fragile point, which is family farming, particularly in the production and support of the food.

Below, we list some lessons learned from the results of this review, and point out alternative ways to mitigate the fragilities found.

*4.1. Challenges We Need to Overcome to Sustain the Supply of Local Food from Family Farming in School Meals*

The inclusion of local foods in school meals has already proven its importance worldwide, but we still need to make progress in consolidating this market. Based on the six barriers identified in this review, we have found three key issues: (1) the lack of public incentives for agricultural/rural development weakens the participation of family farming in school feeding programs; (2) the logistics of the local food supply chain for schools is very sensitive to the actions of actors, markets, and governments; and finally, (3) the quality of school meals depends directly on the success of the previous steps.

First, we consider that without investment in family farming, local foods are unlikely to reach the tables of school children. Nehring et al. [63] believe that support policies for small producers are essential to ensure the success of the participation of family farmers in institutional markets and public programs. We currently observe that the food production and marketing sector is experiencing a great paradox: on the one hand, most of the investments for agricultural development, from public and private sectors, are directed towards export value chains (commodities); while on the other hand, investments for family farming are made by the farmers' own families [64]. Overall, the studies we have gathered indicate that the main difficulty in supplying food for schools is the limited production capacity and irregular supply of food throughout the year. Our experiences have also shown that family farming lacks public policies for agrarian/rural development and financial resources to enter and remain in this school supply market. According to the report Investing in Smallholder Agriculture for Food Security [64], investments in family properties contribute to facilitating producers' access to productive assets (land, inputs, electricity, irrigation, etc.), which allow them to increase their productivity, improve their access to and creation of different markets through strategies that combine public and private investments, and develop state policies that regulate production models and markets suitable for small landowners. Similarly, Birner and Resnick [65] argue that meeting the market demand for family agriculture, with high yields and productivity, requires public strategies to support agricultural production. The authors also argue that institutional markets, for example, are a key political intervention for agricultural development and social protection, as they synergistically stabilize prices, generate income, and ensure food security. Therefore, we believe that increasing investment in family agriculture is the first step in overcoming the limitations of the food supply for school feeding programs. Consequently, this investment is essential for generating income, employment, and means of subsistence, and can be strategic in reducing poverty and food insecurity in rural populations.

In the second place, logistics operation is the most critical step and one that is most susceptible to interference throughout the process of acquiring local foods for schools. This step integrates an extensive and complex network of actors, markets, and governments, and therefore is subject to the influence of various factors. However, most logistical problems fall on the weakest link in this chain, which is the farmer. Studies report several factors that hinder the progress of this step, such as distance, family properties being far from schools and main consumption centers, which makes it difficult to transport fresh food and increases the cost of distribution/delivery [66]; lack of infrastructure, difficulty in storing and transporting their products appropriately [67]; delay in payments, inconsistency in payment by governments weakens negotiations between farmers and school feeding programs [68]; lack of knowledge about legislation regulating programs, difficulty in understanding how to provide food for government programs or other institutions [69]; and weak communication, which hinders understanding of the process and interaction between stakeholders [20]. These problems coincide with the findings of this review. For example, in the United States, Brazil, and Ghana, countries with older and more structured school feeding programs, logistics is a gap that repeats itself in different contexts [68,70,71]. However, the capacity to invest in the family agriculture market, the willingness of govern-

ments (political will), and the degree of engagement of social actors are decisive factors in establishing efficient and well-planned logistics.

Consequently, in the third place, we understand that a lack of incentives for farmers combined with inadequate logistics can negatively impact the quality and availability of food for school feeding programs. The results of this interfere with the quantity and variety of school diets. Studies report an increase in organic foods [72], a decrease in sugar-rich foods [73], and a reduction in ultra-processed [74] foods in school menus when linked to family farming. The supply of biofortified and underutilized foods has also been used as a strategy to promote adequate and healthy nutrition in schools [75,76], especially in low- and middle-income countries. Furthermore, the consumption of locally grown food in schools has been an incentive for building more sustainable food systems [77]. In some Latin American countries, for example, pilot projects known as "Sustainable Schools" are being implemented in school meal programs to include a greater variety of foods from local small farmers [78]. Therefore, we emphasize that ensuring a good quality of school meals involves not only the availability of foo, but a set of measures that support the farmer throughout the entire production, marketing, and distribution chain.

Finally, we highlight that the barriers previously raised can become even more complex depending on the socioeconomic characteristics of each place. Our findings indicate that in low-income countries, for example, the opportunities for farmers to participate in school feeding programs are even more limited. Globally, it is estimated that between 2013 and 2020, the number of children receiving school meals grew by 9% worldwide, reaching a coverage of 388 million [13]. In general, low-income countries have lower coverage, less established programs, and rely more on external funding. On the other hand, high- and middle-income countries stand out for their institutionalized regulatory frameworks, increasing policy support that engages with school feeding, and continuous monitoring of the food and nutrition status of schoolchildren—through population-based food surveys [79,80]. Therefore, we consider that national income level is also a factor that influences the structure of practices linked to family farming in school feeding programs. However, the difficulties discussed in this review need to be contextualized according to local realities, so that future interventions meet the specific needs of each context.

### 4.2. Possible Pathways for Rebuilding the Link between Family Farming and School Feeding

Based on this review, we highlight six main barriers that need to be overcome for the successful engagement of family farming in school feeding programs. We emphasize that the most critical issues arise from the lack of investment in family farming and the inefficient logistics, which can, in turn, impact the quality of school meals. In Box 1, we suggest some strategies to mitigate the difficulties found.

**Box 1.** Strategies to facilitate the supply of food from family farming for school meals.

1. Invest in support policies that improve access for family farmers to markets (institutional or otherwise) and essential public goods.
2. Design laws for the local procurement of school foods with clear objectives, reducing biases for diverse interpretations.
3. Increase investment in family farming and ensuring sustained political commitment to inclusive governance at the local, national, and international levels.
4. Invest in public-private partnerships to offer different market opportunities.
5. Create linkages between farmers and school meal services to strengthen direct and responsive communication.
6. Increase the diversity of school meal menus by including local foods, to ensure better quality meals for students.
7. Strengthen farmer organizations (formal or informal) and improve the logistics of the food supply chain for schools through cooperatives and associations.
8. Invest in national/regional/local research to assess the state of family-origin acquisitions in school feeding programs, from a holistic perspective.

In summary, the suggested strategies recognize that small family farmers need organization, sustainable practices, and social protection. Here we list some arguments that support our proposal. First, the integration of small agricultural producers into food, input, and service supply chains is essential to keep them competitive and protect their means of livelihood. Family farming cooperatives and associations can help enhance this integration by improving the logistics of the supply chain and facilitating access to productive resources, such as machinery, equipment, and rural credit, as well as increasing marketing power [1]. A study conducted in Austrian municipalities identified that cooperatives are vital for small farmers trying to establish themselves in local food supply systems [81]. Cooperatives promoted infrastructure, logistics, and shared transportation, processing methods, as well as agricultural know-how. Experiences in Brazil and the United States have also shown that cooperated farmers have been able to expand food marketing and maintain availability throughout the year, including for school feeding [67,82].

Second, cooperatives can act in the formation of social capital. The linking of social capital within organizations positively influences the establishment of trust, transparency, communication, and commitment among its members and, therefore, can help in overcoming problems [79]. Some studies show the potential of social capital to mitigate family income and food supply shocks, especially in times of crisis [83]. It can be observed that trust and mutual knowledge among members of a social group can increase the propensity for sharing food or money for food purchases [84]. Thus, a lower propensity to hunger is observed in families with higher levels of social capital. Similarly, agricultural productivity can be influenced by social capital, which promotes the exchange of information among its members and the adoption of better agricultural practices and technologies [83]. Therefore, cooperatives and associations are considered potentially effective means of increasing the livelihoods of farmers through the reduction of information asymmetry and transaction costs.

Our third argument is that school feeding programs, which encourage the purchase of local foods, can help minimize the impacts of climate change and the depletion of natural resources. The implementation of shorter, local food supply chains can have several benefits. In addition to promoting a preference for fresh foods, this strategy can also help reduce the need for transportation, which leads to lower carbon emissions [13]. Furthermore, supporting local small farmers can increase the resilience of local food systems. However, strategies to support family farming should consider the adverse effects of climate change [85]. According to the Special Report on Climate Change and Land [86], farmers are particularly vulnerable to climate change because their livelihoods often depend on agricultural production. As extreme weather events become more frequent, producers will need subsidies that provide immediate and short-term relief in cases of agroclimatic disasters [1]. However, long-term measures are necessary to strengthen the resilience of farming families [87].

Overall, we believe that school feeding programs can be even more effective when approached from a holistic and multisectoral perspective that takes into account biological, social, cultural, economic, and environmental dimensions. These programs are typically associated with education and health agendas, where efforts are focused on combating child hunger and nutritional deficiencies and increasing school participation and learning. However, evidence shows that the potential benefits of these programs extend to at least four main sectors: health, education, social protection, environmental protection, and agriculture [88]. The effects of this "win-win" relationship cross sectoral boundaries and impact various domains. For example, from a nutritional and agricultural perspective, biofortified foods can be incorporated into school meals, bringing health benefits as new technologies are developed and local agricultural production is maintained [89]. From an economic perspective, the programs can have a good cost-benefit ratio when viewed from the perspective of their multisectoral return, which can reach up to USD 9 in benefits for every USD 1 invested, meaning that school feeding can generate returns on investment in other sectors [88]. In terms of education and social protection, school feeding also reduces

school dropout, which in turn improves educational performance and reduces the risk of child labor [89]. For example, Dyngeland et al. [90] analyzed the Zero Hunger strategy in Brazil and found that investments in the National Program for Strengthening Family Agriculture (PRONAF) and the rural credit program were strategic in driving the success of social policies, especially in synergy with the SDGs. Thus, the investments in Zero Hunger were translated into advances to increase the availability of food (SDG 2), reduce poverty (SDG 1), and conserve natural vegetation (SDGs 13 and 15).

## 5. Conclusions

Problems in linking family farming to school feeding programs are most evident in food production, where reduced productive capacity, irregular supply, and inefficient logistics are more common. Barriers, such as current local legislation, financial costs, and lack of communication, also hinder the relationship between farmers and program-responsible parties. The lack of locally grown foods also affects diversity and, consequently, the quality of school menus. The lack of incentives and support for family farming, lack of access to markets, and fragile logistics operations, including marketing, distribution, transportation, and delivery, are the main causes of the difficulties encountered in acquiring local foods for school feeding programs.

Additionally, we highlight that the inclusion of farmers in cooperatives and associations can be an effective solution to improve the logistics infrastructure of the supply chain for schools. However, for this option to be viable as a market alternative, more financial and political incentives for the family farming sector are necessary. In this way, the supply of local foods for national programs becomes a more attractive option for farmers.

It must be considered that in our analysis there is a higher concentration of studies related to the Farm to School Program and PNAE, as we understand that these programs are older, more structured, and successful. Thus, there is a lack of diversity of experiences in this approach and also a gap in publications that portray these scenarios, especially in low-income countries. Furthermore, we suggest more empirical studies that evaluate the impact of these practices on national programs and examine the logistics of the supply chain in schools. We hope that our study can shed light on the limitations surrounding the inclusion of local foods in national school feeding programs and contribute to the reformulation of agricultural and food policies that aim to increase the participation of farmers in this market niche.

**Supplementary Materials:** The following supporting information can be downloaded at: https://www.mdpi.com/article/10.3390/su15042863/s1, File S1: PRISMA Checklist; File S2: Research strategy for systematic review.

**Author Contributions:** V.M.C., S.M.G. and M.C.M.J. participated in all phases of the study. C.R. participated in the study analysis, writing and critically reviewing the intellectual content. J.B.A.d.C. contributed to the writing and revision of the manuscript. All authors have read and agreed to the published version of the manuscript.

**Funding:** This research received no external funding.

**Institutional Review Board Statement:** Not applicable.

**Informed Consent Statement:** Not applicable.

**Data Availability Statement:** All relevant data are within the manuscript and its Supplementary Materials.

**Conflicts of Interest:** The authors declare no conflict of interest.

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
