# Peer review of "Integrating Family Farming into School Feeding: A Systematic Review of Challenges and Potential Solutions"

_sustainability, doi:10.3390/su15042863_

Round 1
Reviewer 1 Report
The article is well structured. The relevance of the problem has been clarified. The formulated objective is clear and indicates the scope of the research.
The methodology describes in detail the steps of the analysis of a wide range of literary sources on the researched topic (figure 1).
The results of the analysis are presented in table 1 - characteristics of studies on family farming in school feeding.
The authors identify eight strategies with the potential to improve school feeding by providing food from local farms.
Author Response
Dear Reviewer,
Thank you for your review of my manuscript entitled "Integrating family farming into school feeding: a systematic review of challenges and potential solutions" I appreciate your valuable feedback and suggestions for improvement.
Thank you again for your time and consideration.
Sincerely,
Reviewer 2 Report
The manuscript is innovative in addressing food systems matters.
Although, the review is very rich and could be confusing and but the clearness and organization of the manuscript made it easier yet comprehensive.
Almost all related details were covered in the main manuscript and in the supplementary files.
It would be better to:
1- endorse by a reference the rational behind why an explanatory review can be not limited in the year or design (mentioned in the abstract + L98-99).
2- explain why reports and theoretical essays were excluded from the review (L99-100)
3- clarify why 66 articles were selected for full reading based on the titles and abstracts (provide more info about this selection, ex: which titles, etc.)
Author Response
Dear Reviewer,
Thank you for your review of my manuscript entitled "Integrating family farming into school feeding: a systematic review of challenges and potential solutions" I appreciate your valuable feedback and suggestions for improvement. We respond to your comments below:
Comments and Suggestions:
The manuscript is innovative in addressing food systems matters. Although, the review is very rich and could be confusing and but the clearness and organization of the manuscript made it easier yet comprehensive. Almost all related details were covered in the main manuscript and in the supplementary files. It would be better to:
1- endorse by a reference the rational behind why an explanatory review can be not limited in the year or design (mentioned in the abstract + L98-99).
- Answer: Our systematic review was designed as an exploratory study to investigate the problems that school feeding programs in different contexts face in acquiring food from family farming. In order to answer this question in the most comprehensive way possible, it was important to include as many relevant studies as possible. We believe that by not limiting the year, we were able to conduct a complete and comprehensive review of the existing evidence on the subject. Limiting the year of the studies in the search would result in the exclusion of studies that, although older, can still provide valuable information and contribute to our understanding of the subject.
- We added this explanation in a more brief way to the manuscript. See p.3, Line 104-106 and in the abstract.
2- explain why reports and theoretical essays were excluded from the review (L99-100)
- Answer: In our review, we focused only on original research that collected data through direct observation or experimentation. While theoretical essays and reports may provide valuable perspectives and background information, they do not provide primary data and in some cases may not be subject to the same level of rigor as primary studies. Therefore, we made the decision to exclude these types of studies to ensure a comprehensive and unbiased evaluation based on existing evidence on the topic. Additionally, it was important to us to ensure that our review would not be influenced by the authors' own bias or interpretation. By including only primary studies, we were able to minimize potential bias in our review.
- This explanation has been added in a more concise form to the manuscript. See p.3, Line 106-108.
3- clarify why 66 articles were selected for full reading based on the titles and abstracts (provide more info about this selection, ex: which titles, etc.)
- Answer: Our search of studies in databases generated a much larger number of articles than those that are actually eligible by the eligibility criteria. This occurred because the search strategy is designed to prioritize sensitivity over specificity. Thus, a reading of the titles and abstracts in this first phase allowed us to perform a screening of these references and discard a large number of references that do not fit the criteria established by our review. The titles and abstracts were selected from the descriptors used in the search strategy (see in "Supplementary Material 2"). So, 66 articles passed in this first screening to be read in full.
- We improved this explanation in the "3. Studies selection" topic. See page 3, lines 112-119.
Thank you again for your time and consideration.
Reviewer 3 Report
This review gathered the evidence on the implementation of family farming connections to school food systems. A wide and broad search strategy and coding approach was taken to gather and code the existing evidence. The authors did a particularly excellent job of acknowledging the diversity of farming internationally, while maintaining a concise definition for the purposes of the work. This helps the readers both understand the framing and also apply the results to their own context. My major concern relates to the lack of details on how themes and conclusions were reached. While you describe the coding of study quality in detail, there is very limited description of how you identified themes (e.g. Figure 3; Box 1). In addition, your research question includes only "problems" but you include significant discussion of "solutions" in the Discussion. I suggest expanding your RQ so that those results make more sense. Some other smaller comments follow.
Methods:
- The authors use the word "exploratory" to describe their review in several places, despite using high quality systematic review methodology. I would either clarify what you mean by exploratory, or remove that descriptor.
Results:
- Define "low quality" for the purposes of exclusion (line 148)
Author Response
Dear Reviewer,
Thank you for your review of my manuscript entitled "Integrating family farming into school feeding: a systematic review of challenges and potential solutions" I appreciate your valuable feedback and suggestions for improvement. We respond to your comments below:
Comments and Suggestions:
This review gathered the evidence on the implementation of family farming connections to school food systems. A wide and broad search strategy and coding approach was taken to gather and code the existing evidence. The authors did a particularly excellent job of acknowledging the diversity of farming internationally, while maintaining a concise definition for the purposes of the work. This helps the readers both understand the framing and also apply the results to their own context. My major concern relates to the lack of details on how themes and conclusions were reached. While you describe the coding of study quality in detail, there is very limited description of how you identified themes (e.g. Figure 3; Box 1). In addition, your research question includes only "problems" but you include significant discussion of "solutions" in the Discussion. I suggest expanding your RQ so that those results make more sense. Some other smaller comments follow.
- Answer: Done. We have made adjustments to our research question. See page 2, Line 81-82. As for the categorization of our results, we carried out a detailed systematization of the main conclusions of the studies included in this review. We grouped the identified problems into themes that were similar and based on that we made the categorization, which can be seen in Figure 3. We provided a brief explanation in the section "5. Summary of Results". See page 4, Line 154-157.
Methods:
- The authors use the word "exploratory" to describe their review in several places, despite using high quality systematic review methodology. I would either clarify what you mean by exploratory, or remove that descriptor.
- Answer: Done. We remove the descriptor. See in the abstract and p.3, Line 104-105.
Results:
- Define "low quality" for the purposes of exclusion (line 148)
- Answer: We described how we classified studies as "low quality" in the section "2.4. Data extraction" 2nd and 3rd paragraph. We improved the writing of this information in the section "3.1. Studies included." See p. 4, Line 165-166.
Thank you again for your time and consideration.
Reviewer 4 Report
I recommend publishing. As someone who has worked directly with research and practitioners on these programs, it was a very helpful overview of the findings across multiple studies. It was clear, the writing flowed well, the table was well structured and analysis easy to follow. Its contribution is significant in terms of the usefulness of this work across the field of researchers studying the use of local foods in school meal programs. The only suggestion I have for the authors is to note the heavy reliance on research of the US Farm to School program, and the absence of a discussion on any research on informal or even formal networks that provide local farm food to schools if more countries were included. I am not suggesting this be part of this particular article, but that the article at least raise the issue of the lack of diversity in research on the topic globally, and, if the authors agree, suggesting this as a research gap that could be addressed by future scholars.
Author Response
Dear Reviewer,
Thank you for your review of my manuscript entitled "Integrating family farming into school feeding: a systematic review of challenges and potential solutions" I appreciate your valuable feedback and suggestions for improvement. We respond to your comments below:
Comments and Suggestions:
I recommend publishing. As someone who has worked directly with research and practitioners on these programs, it was a very helpful overview of the findings across multiple studies. It was clear, the writing flowed well, the table was well structured and analysis easy to follow. Its contribution is significant in terms of the usefulness of this work across the field of researchers studying the use of local foods in school meal programs. The only suggestion I have for the authors is to note the heavy reliance on research of the US Farm to School program, and the absence of a discussion on any research on informal or even formal networks that provide local farm food to schools if more countries were included. I am not suggesting this be part of this particular article, but that the article at least raise the issue of the lack of diversity in research on the topic globally, and, if the authors agree, suggesting this as a research gap that could be addressed by future scholars.
- Answer: We added in the conclusions this observation regarding the gap of studies with more global experiences. See p. 26, Line 490-494.
Thank you again for your time and consideration.
Round 2
Reviewer 3 Report
Thank you for your revisions and changes, which strengthened the quality of the manuscript. Information regarding data analysis and theme generation needs to be provided in the methods section, as it relates to the way data were collected and analyzed. Ideally, this would include a qualitative data analysis framework, if used. At the moment, the validity of the results in the qualitative portion (e.g. solutions and problems) cannot be assessed. The info provided in lines 194-206 begin to present this information, but more detail is needed and this discussion belongs in the methods section.
Author Response
Dear Reviewer,
Thank you for your review of my manuscript entitled "Integrating family farming into school feeding: a systematic review of challenges and potential solutions". I appreciate your valuable feedback and suggestions for improvement. Below we answer your comment.
Answer: Done. We have added further explanation of how we performed the qualitative data analysis as requested. We added this information in the topic “2.5. Summary of results”, located in the study methods. See p. 4, Line 154-163.
Thank you again for your time and consideration.
Sincerely,
Round 3
Reviewer 3 Report
The additions to the Methods section addressed my previous concerns and added significant value to the manuscript, allowing readers to understand the theme generation process.